# How to Confuse Motor Control: Passive Muscle Shortening after Contraction in Lengthened Position Reduces the Muscular Holding Stability in the Sense of Adaptive Force

**DOI:** 10.3390/life13040911

**Published:** 2023-03-30

**Authors:** Frank N. Bittmann, Silas Dech, Laura V. Schaefer

**Affiliations:** 1Regulative Physiology and Prevention, Department Sports and Health Sciences, University of Potsdam, 14476 Potsdam, Germany; 2Sports Education, Department Sports and Health Sciences, University of Potsdam, 14476 Potsdam, Germany

**Keywords:** maximal isometric Adaptive Force, holding capacity, muscle stability, neuromuscular functioning, neuromuscular control, motor control, muscle spindle, muscle physiology, regulatory physiology

## Abstract

**Simple Summary:**

During everyday activities and sports, muscles must continuously stabilize the body. In particular, to change direction or to decelerate in motion requires well-dosed muscular resistance. Accordingly, muscle length and tension need to be adjusted. Complex control processes of the nerve–muscle system are demanded. To hold a limb stable under the load of varying forces, the muscle-controlling system has to organize matching resistance. It was previously shown that the holding capacity is sensitive to interfering inputs such as unpleasant odors or imagery. We investigated if different preconditioning regarding contraction and length changes of elbow flexors (*n* = 19 limbs) influence the maximal holding capacity and maximal Adaptive Force. Muscles were manually tested using a handheld device (recording force and limb position). A passive shortening of the muscle after it was contracted in the lengthened position caused a breakdown of the holding capacity by ~47%. The maximal Adaptive Force was not affected. A second precontraction in the test position erased the reducing effect. This passive shortening is assumed to lead to a slack of muscle spindles (receptor for muscle length). This presumably causes inappropriate length information and irritates muscle length and tension control in the short term. The results reflect the functional characteristics of the muscular holding capacity in contrast to other strengths.

**Abstract:**

Adaptation to external forces relies on a well-functioning proprioceptive system including muscle spindle afferents. Muscle length and tension control in reaction to external forces is most important regarding the Adaptive Force (AF). This study investigated the effect of different procedures, which are assumed to influence the function of muscle spindles, on the AF. Elbow flexors of 12 healthy participants (*n* = 19 limbs) were assessed by an objectified manual muscle test (MMT) with different procedures: regular MMT, MMT after precontraction (self-estimated 20% MVIC) in lengthened position with passive return to test position (CL), and MMT after CL with a second precontraction in test position (CL-CT). During regular MMTs, muscles maintained their length up to 99.7% ± 1.0% of the maximal AF (AF_max_). After CL, muscles started to lengthen at 53.0% ± 22.5% of AF_max_. For CL-CT, muscles were again able to maintain the static position up to 98.3% ± 5.5% of AF_max_. AFiso_max_ differed highly significantly between CL vs. CL-CT and regular MMT. CL was assumed to generate a slack of muscle spindles, which led to a substantial reduction of the holding capacity. This was immediately erased by a precontraction in the test position. The results substantiate that muscle spindle sensitivity seems to play an important role for neuromuscular functioning and musculoskeletal stability.

## 1. Introduction

The term proprioception, presumably first named in 1882 by Ardigo [1], includes the senses of tension, force, effort, and balance, as well as the senses of limb position and movement [2]. The term kinesthesia, introduced by Bastian in 1888 [3], refers to the latter two sensations [2]. Proske and Gandevia argued that “muscle spindles play the major role in kinesthesia, with some skin receptors providing additional information.” [2]. It is known that muscle spindles provide information to the central nervous system, e.g., muscle length and limb position, changing status of muscle tone and movement [4]. An elaborate overview on the current neurophysiological knowledge about muscle spindles was given by Macefield and Knellwolf [5]. Most investigations regarding muscle spindles considered the behavior under conditions such as stretching or contraction [5,6,7,8,9,10]. Meanwhile, it is commonly accepted that a slack in muscle spindles can occur [11,12]. This was based on the early suggestion that “… following movements, the intrafusal fiber will always form stable bridges; when this is at a long length, the fiber on returning to its rest length will be (stiff but) slack…” [13]. Blum et al. presented a mathematical model in 2020 which underpins the experimental assumptions by providing information that, under particular conditions, a slack of spindle cells could appear [14]. The ability of muscle fibers to fall slack is considered as one important consequence of the thixotropic behavior of muscles [8,15,16]. Shortened or slacked muscle spindles are mainly examined in terms of reflex reactions [2,5,17,18,19,20]. The research group around Proske introduced a procedure in humans which is suggested to generate a slack in muscle spindles: “… if the muscle is stretched, contracted at the stretched length, and held there for several seconds, stable cross-bridges will form at the longer length… On return to the initial length, the intrafusal fibers, stiffened by the stable cross-bridges, are unable to shorten themselves and fall slack” [15]. They showed that this lowered the background discharge rate [15,21]. Gregory et al. found that the stretch reflex was significantly reduced by this procedure, specifically, when a muscle was passively brought into middle length (test position) after it was contracted briefly in a lengthened position with 25% of the maximal voluntary isometric contraction (MVIC) [18]. The procedure prior to the reflex test was assumed to lead to a slack in muscle fibers, both extrafusal and intrafusal [18]. Under such conditioning, the muscle spindles provide sensory information which are not in line with the overall muscle length. This was discussed to lower the reflex intensity [18]. The phenomenon of a reduced reflex following such a “slack procedure” was immediately revoked by a short submaximal voluntary contraction (2 s at 5%, 10%, or 25% of MVIC) at the test length. Hereinafter, the reflex response increased significantly, especially for a precontraction of 10% of MVIC. For 25% of MVIC, no further significant increase occurred. It was assumed that the slacked muscle spindles get tightened again by that second contraction in test length [18]. Applying a percutaneous electrical muscle stimulation (assumed to be solely extrafusal) instead of the voluntary second contraction did not lead to this normalization of reflex activity. Therefore, it was assumed that a voluntary fusimotor activation of at least 10% of MVIC is necessary to fully remove the slack in muscle spindles [18]. Héroux and colleagues showed for vastus lateralis muscle that already 5% of MVIC was sufficient to reduce muscle slack length [20]. Meanwhile, it is common knowledge that the reflex behavior of muscles can change depending on the previous history of contraction and length changes.

Although investigations on such particular preconditioning of muscle fibers mostly use reflex behavior as the target parameter, it is conceivable that other motor functions can also be affected. The sense of force, for example, was found to not react to a preceding contraction in lengthened position [22]. To our best knowledge, there are no studies which investigated muscle strength after such preconditioning, except for studies regarding common stretching exercises [6,10,23], which clearly differ from the procedure meant here. Investigations on the effect of conditioning procedures as mentioned before on muscle strength are indicated. Since muscle spindles are especially relevant for length control, the effect of a presumed slack of muscle spindles on adaptive capabilities in the sense of length and tension control in reaction to external forces is especially relevant.

A promising new approach to assess the adaptability of the sensorimotor system is the Adaptive Force (AF). “AF not only requires muscle strength but also sensorimotor control. It reflects the neuromuscular functionality to adapt adequately to external forces with the intention of maintaining a desired position or movement” [24]. An appropriate adaptation to external forces provides dynamic stability of the musculoskeletal system. Adaptation to external forces doubtlessly relies, inter alia, on a well-functioning proprioceptive system including muscle spindle afferents. The latter provide information about muscle length and its change, which is inevitable for appropriate motor responses to external impacts. Hence, it was questioned whether a preconditioning procedure such as the one mentioned above [15,18] would affect the AF and, thus, the musculoskeletal stability. In case the stabilizing holding capacity of muscles would be impaired, instability and, thus, risk of injury could be a consequence. Despite an abundance of investigations, the occurrence of muscle or tendon injuries in sports but also in everyday life without trauma or other comprehensible causation is still an unsolved enigma. The detailed mechanisms of such injuries remain unclear. Researchers mainly agree that damages mostly occur during non-contact actions while muscles try to decelerate external loads [25,26]. It was suggested that a crucial mechanism could be “when active muscular restraints are unable to adequately reduce joint torques during dynamic movements involving deceleration and high forces” [27,28,29]. Hence, investigating the AF and its reactions to preconditioning, which presumably alters muscle spindle afferents, might also be a beneficial approach regarding further information on injury mechanisms.

This study investigated the behavior of AF of elbow flexors—as an approach to describe musculoskeletal stability and neuromuscular functioning—after two different procedures assumed to influence muscle spindles. The crucial parameter of AF is the maximal isometric AF (maximal holding capacity; AFiso_max_) which characterizes the maximal force under which a muscle is able to stabilize a given limb position (isometric condition) against an increasing external force. In case the muscle starts to lengthen during the external force increase (exceeding AFiso_max_), the force usually increases further during the subsequent eccentric action until the maximal AF (AF_max_) is reached.

It was shown previously that AFiso_max_ was significantly reduced under disturbing conditions such as unpleasant smell or imagery in healthy participants, whereby AF_max_ stayed on the baseline level [30,31,32]. Recently, it was shown that the holding capacity was significantly reduced in patients with long COVID and stabilized with recovery [33,34]. The AF_max_ again did not show this behavior and was on a considerably high level already in the long COVID state. Hence, the maximal holding capacity (AFiso_max_) seems to be especially sensitive to inputs.

On the basis of these findings, the following was hypothesized:(1)AFiso_max_ would be significantly reduced by a brief submaximal precontraction in a lengthened position followed by a passive return to test position (procedure CL), whereas AF_max_ would not be affected.(2)An additional brief precontraction in the test position (directly after procedure CL, procedure CL-CT) would immediately revoke the reduction of the holding capacity.

The study gains novel insights regarding the behavior of musculoskeletal stability and neuromuscular functioning in the sense of AF after specific procedures, which are assumed to manipulate the function of muscle spindles. As part of muscle physiology, this is relevant for the understanding of neuromuscular control during stabilizing actions and, hence, might provide further insights into injury mechanisms.

## 2. Materials and Methods

Measurements were conducted at one appointment at the Neuromechanics Laboratory of the University of Potsdam (Potsdam, Germany). One experienced male tester (65 years old, 185 cm, 87 kg, 27 years of manual muscle test (MMT) experience) performed all tests of the elbow flexor muscles, which were objectified using a handheld device.

### 2.1. Participants

G*power (version 3.1.9.7, Düsseldorf, Germany) was used for a priori sample size estimation. For the main comparison regarding the different procedures (regular MMT vs. CL vs. CL-CT procedures), within-factor RM ANOVA was chosen (α = 0.05, 1 − β = 0.8, correlation: 0.5, non-sphericity correction: 1, number of groups: 1, number of measurements: 3). To determine a substantial effect size of f = 0.4, a minimum of *n* = 12 participants was revealed. According to previous studies investigating AF parameters using MMTs [30,31,32], the effect sizes were very large; hence, the effect size of 0.4 estimated herein can be considered conservative.

Twelve healthy participants (m = 9, f = 3) volunteered to participate in this study. Females were 31.33 ± 6.81 years old (range: 26–39), weighed 56.33 ± 3.79 kg, and were 170.00 ± 6.00 cm tall; corresponding values for males were 30.89 ± 9.23 years (range: 22–52), 79.22 ± 12.42 kg, and 184.33 ± 8.92 cm, respectively. Exclusion criteria were any current orthopedic diagnosis, injuries, and/or surgery of cervical spine and upper extremities within the last 12 months. Furthermore, an affected neuromuscular function of elbow flexors, assessed by an MMT in the sense of a break test (see below) prior to the measurements, led to exclusion, as did any complaints during its assessment. Elbow flexors of both sides were examined. Four participants could only be measured on one side due to an affected neuromuscular control. One further participant needed to leave after measurements of the first side. In total, 19 limbs were examined and considered for evaluation (*n* = 12 left, *n* = 7 right).

The study was conducted according to the guidelines of the Declaration of Helsinki and was approved by the Ethics Committee of the University of Potsdam, Germany (protocol code 35/2018; 17 October 2018). All participants gave their written informed consent to participate.

### 2.2. Technical Equipment

A wireless handheld device (Figure 1a) was used to record the reaction force (dynamometry) and angular velocity (gyrometry) during the MMTs [30,31,32,33,34,35]. It combines strain gauges (co. Sourcing map, model: a14071900ux0076, precision: 1.0% ± 0.1%, sensitivity: 0.3 mV/V) and kinematic sensors (Bosch BNO055, 9-axis absolute orientation sensor, sensitivity: ±1%). Data were recorded with a sampling rate of 180 Hz, AD-converted, and transmitted via Bluetooth to a tablet (Sticky notes, comp: StatConsult, Magdeburg, Germany).

### 2.3. Manual Muscle Tests

The AF was examined by means of a MMT in the sense of a break test. This has been described previously [30,31,32,33,34,35,36]. In brief, the task of the participant was to maintain the static test position despite an external increasing force applied by the tester. Hence, the participant had to perform a holding isometric muscle action (HIMA) [37,38,39]. The tester applied the external force rise by pushing against the participant’s limb in the direction of muscle lengthening. The suggested optimal force profile (S-shaped) was described in detail by Bittmann et al. [35]. This force profile needs to be reproducible to achieve valid empirical data [35]. The tester of this study proved his ability to test reproducibly (CV = 4.6%; ICC = 0.995) [35]. Basically, two qualities during MMT could arise [30,31,32]. Firstly, if the participant was able to maintain the static starting position during the entire force increase applied by the tester, the MMT was rated as “stable”. This does not per se test the maximal holding force of the participant, since the applied force under stable conditions depends on the tester’s maximal force. The tester was instructed to increase the force up to a considerably high intensity. If the tester’s pushing force exceeded the maximal holding capacity of the participant, the limb gave way during the force increase, and the MMT was rated as “unstable”. The force usually increases further during the subsequent eccentric phase until the maximal AF (AF_max_) is reached.

Before and after the measurements using the MMT in the sense of a break test to assess the AF (named MMT hereafter), the MVIC was assessed by means of a make-test [36]. Actually, this is also a manual muscle test. For better differentiation, it is named MVIC test hereafter. For this, the participant had the task of pushing as strongly as possible against the handheld device which was placed in the palm of the tester who just provided a stable resistance. Hence, the participant performed a pushing isometric muscle action (PIMA) [37,38,39].

### 2.4. Setting and Measurement Procedure

The setting for measuring elbow flexor muscles is shown in Figure 1b. The participant was placed supine. The starting (test) position of all tests was a flexed elbow joint in 90° and a maximally supinated forearm. The tester placed the handheld device in his palm and contacted the distal part of the forearm. For standardization of the test position, the contact point at the forearm was marked.

Two MVIC tests and two MMTs without any procedure (regular MMT) were performed for reference—one before and one after the other MMTs with procedures for each. For that, the participants took up the test position by themselves. Between those MVIC and regular MMTs four MMTs were conducted after two different procedures:*Procedure CL: precontraction in lengthened position with passive return*From the test position, the elbow joint was brought passively into maximal extension by the tester (neutral zero position with maximal supination of the forearm). In that position, the participant was instructed to push shortly (~1 s) with self-estimated 20% of the MVIC against a stable resistance, which was provided by the tester. The handheld device recorded the force of precontraction between the tester’s palm and participant’s forearm. Afterward, the tester guided the limb back to the test position of the MMT. To ensure that the elbow flexors stayed passively and did not support the flexion actively, the participant pushed slightly against the tester (activation of elbow extensors) during the return. Due to the passive shortening after precontraction in the lengthening position, this procedure was assumed to produce a slack in muscle fibers. Back in the test position after the CL procedure, the tester started the MMT after ~2 s to achieve a temporal sequence similar to the second procedure.*Procedure CL-CT: CL with subsequent second precontraction in test position*Procedure CL was extended by a second precontraction immediately after the forearm was returned to the test position. The second contraction also amounted to self-estimated 20% of the MVIC and lasted ~1 s. Immediately after this second precontraction, the MMT was performed to assess the AF. It was assumed that procedure CL-CT eliminates the slack in muscle fibers. A minimal intensity of 10% of the MVIC was regarded as necessary to resolve the reflex activity [18]. Hence, 20% of the MVIC was chosen to ensure that this minimal level would be certainly achieved (considering that the self-estimation would show some variance).
Both procedures were practiced before measurements and were accompanied by commands starting from the test position: “stay relaxed” (forearm was positioned passively into neutral zero position with maximal supination); “contract” (short contraction (1 s) with ~20% of MVIC); “stop” (contraction was released); “stay relaxed and push” (passive repositioning of the forearm to 90° elbow flexion with slight activity of elbow extensors); either “stay relaxed” (wait for 2 s) and “hold” (MMT started) for procedure CL or “contract” (second short contraction), “stop” (release contraction), and “hold” (MMT started) for procedure CL-CT. Both procedures were alternately conducted twice.

In total, eight trials were performed: (1) MVIC test, (2) regular MMT, (3) MMT after CL, (4) MMT after CL-CT, (5) MMT after CL, (6) MMT after CL-CT, (7) regular MMT, and (8) MVIC test. The tester rated the stability of the participant’s resistance during the MMTs as “stable” or “unstable”.

### 2.5. Data Processing and Statistical Analyses

The evaluation was performed according to Schaefer et al. [30,31,32,34]. Force and gyrometer signals were analyzed using DIAdem 2017 (National Instruments, Austin, TX, USA). To ensure equidistant time channels, signals were interpolated (linear spline, sampling rate: 1000 Hz). Furthermore, a Butterworth low-pass filter was applied (cutoff frequency 20 Hz, filter degree 5). The following parameters were extracted:MVIC: the peak value of each MVIC test was determined. The peak value of the first MVIC test referred to the individual’s MVIC. The second MVIC test was analyzed to investigate possible fatiguing effects in comparison to the initial MVIC. According to the gyrometer signals, all MVIC tests were conducted under static conditions.Maximal Adaptive Force (AF_max_): the peak value of each MMT trial was selected and referred to the AF_max_ of a single MMT. This was either reached during isometric actions (stable MMT) or during eccentric ones (unstable MMTs). For the former, AF_max_ = AFiso_max_. For the latter, AF_max_ > AFiso_max_ (Figure 2).Maximal isometric Adaptive Force (AFiso_max_): this refers to the highest force value under isometric conditions during the MMT. The gyrometer signal was used to determine if the forearm moved in the direction of elbow extension during the force increase (breaking point), indicating muscle lengthening. If muscle lengthening occurred, the force value at the breaking point referred to AFiso_max_ (Figure 2a). In case the static position was maintained up to the peak value, AFiso_max_ = AF_max_ (Figure 2b). For a detailed description, see [30,31,32,34].Adaptive Force at the onset of oscillations (AFosc): this refers to the force at the moment in which a clear upswing of subsequent oscillations in the force signal appeared (for details, see [30,31,32,34]) (Figure 2b). In case a clear upswing did not occur, AFosc = AF_max_ (Figure 2a).Slope: the difference quotient was used to determine the slope before the breaking point. Reference points were time and force of 70% and 100% of the averaged AFiso_max_ of all MMTs of one muscle assessed as unstable. The decadic logarithm was taken from slope values since the slope rise was exponential [lg(N/s)].

Additionally, the ratios $\frac{{\mathrm{AFiso}}_{\max}}{{\mathrm{AF}}_{\max}}$, $\frac{\mathrm{AFosc}}{{\mathrm{AF}}_{\max}},\;\mathrm{and}\;\frac{{\mathrm{AFiso}}_{\max}}{{\mathrm{AF}}_{\mathrm{osc}}}$ were calculated (%).

To compare the three precontractions (first of procedure CL, and first and second of procedure CL-CT), the averaged force (related to MVIC (%)) and the duration (s) of the isometric plateaus were captured (Figure 2). Interval borders for the isometric plateaus were determined by setting 90% of the peak value of the related precontraction as reference. Furthermore, the duration (s) from the end of first precontraction to the start of MMT was compared between both procedures (CL vs. CL-CT) (Figure 2).

The arithmetic means (M), standard deviations (SD), and 95% confidence intervals (CI) of all parameters were calculated for regular MMT, MMT after CL, and MMT after CL-CT. Since AF_max_ values of regular MMTs before and after MMTs with procedures did not differ significantly (t (18) = 0.777, *p* = 0.447, two-tailed), they were considered together. Two trials of different participants (first MMT after procedure CL and first MMT after procedure CL-CT) were excluded due to technical issues and elbow pain during measurement.

For statistical evaluation, SPSS Statistics 27 (Windows, Version 28.0. Armonk, NY, USA: IBM Corp.) was used. Main comparisons were the analyses of differences in regular MMT, MMT after procedure CL, and MMT after procedure CL-CT for all parameters. The normal distribution was checked by Shapiro–Wilk test. Because repeated-measures ANOVA (RM ANOVA) is considered to be robust against violation of normal distribution [40,41], it was also used if not all groups were normally distributed. This was the case for $\frac{{\mathrm{AFiso}}_{\max}}{{\mathrm{AF}}_{\max}}$ and slope (each two groups were not normally distributed). If sphericity was not fulfilled (Mauchly test: *p* < 0.05), the Greenhouse–Geisser correction was applied (F_G_). The effect size partial eta squared (η^2^) was estimated by SPSS. For pairwise comparisons, Bonferroni correction was applied (*p*_adj_), and the effect size Cohen’s *d_z_* was interpreted as “small” (0.2), “moderate” (0.5), “large” (0.80), or “very large” (1.3) [42].

Additionally, the MVIC before vs. after MMT trials and the duration from the end of the first precontraction to the start of MMT (comparability between procedures CL and CL-CT) were compared by paired *t*-test (two-tailed). The effect size was given by *d_z_*. The significance level was set at α = 0.05.

## 3. Results

### 3.1. Precontractions: Duration and Force

The values of precontraction phases for procedures CL and CL-CT are given in Table 1. RM ANOVA revealed no significant differences between the relative forces of precontractions (F_G_ (1.23, 22.16) = 2.942, *p* = 0.094). The duration of precontraction phases revealed a significant main effect (F (2, 36) = 17.901, *p* < 0.001, η^2^ = 0.499). Pairwise comparisons showed a significantly shorter second precontraction for procedure CL-CT vs. the first precontraction of both procedures (*p*_adj_ < 0.001 for both; CL vs. second contraction of CL-CT: *d_z_* = 1.143; first vs. second contraction of CL-CT: *d_z_* = 1.190). Generally, the self-estimated amount of precontraction ranged from 10.30% to 44.46% of the MVIC for CL, and from 11.51% to 46.91% and from 14.82% to 43.48% for the first and second precontractions of CL-CT, respectively. Hence, the required minimal force for precontraction was reached for all trials. Furthermore, the duration from the end of the first precontraction to the start of MMT did not differ significantly for both procedures (t (18) = 0.325, *p* = 0.650). Hence, the precontraction conditions can be regarded as similar between both procedures as a prerequisite for further considerations.

### 3.2. Parameters of Adaptive Force in Comparison of the Different Procedures

The values of all AF parameters are given in Table 2. The requirement for a valid comparison between the procedures was given, since the slope of force increase did not differ significantly between procedures CL and CL-CT (*p*_adj_ = 0.180). However, the slope of regular MMT was significantly lower compared to procedures CL (*p*_adj_ < 0.001, *d_z_* = 1.662) and CL-CT (*p*_adj_ = 0.002, *d_z_* = 0.977).

AF_max_ was statistically similar comparing all procedures (regular, CL, and CL-CT) (Table 2, Figure 3a), although close to significance. The lowest value was found for MMT after procedure CL-CT, while the highest was found for MMT after procedure CL.

For AFiso_max_, RM ANOVA was found to be significant (Table 2). Pairwise comparisons revealed a significantly lower AFiso_max_ for procedure CL compared to regular MMT (*p*_adj_ < 0.001, *d_z_* = 1.809) and to procedure CL-CT (*p*_adj_ < 0.001, *d_z_* = 1.434). AFiso_max_ did not differ significantly between regular MMT and procedure CL-CT (*p*_adj_ = 0.555) (Figure 3b). By contracting the muscle in lengthened position with ~25% of the MVIC and a passive return to the test position (procedure CL), AFiso_max_ was clearly reduced by −45.10% ± 25.18% compared to regular MMT. In case a second slight precontraction was performed in middle-length test position (procedure CL-CT), AFiso_max_ was similar to the value reached during regular MMT (−3.81% ± 12.37%).

Figure 4 depicts the single values of the ratio AFiso_max_ to AF_max_ of each of the 19 tested limbs for both MMT trials after procedure CL and CL-CT, respectively. It is visible that, for CL-CT, the ratio amounted ~100% with two exceptions showing lower values (limbs no. 8 and 9, Figure 4). In contrast, clearly lower ratios were obtained for CL with one exception (limb no. 19, Figure 4). Accordingly, RM ANOVA was highly significant (Table 2). Pairwise comparisons revealed significantly lower ratios for CL compared to regular MMT (*p* < 0.001, *d_z_* = 2.098) and compared to CL-CT (*p* > 0.001, *d_z_* = 1.956) (Figure 3d).

This indicates that a precontraction in lengthened position followed by a passive return to middle-length test position immediately reduced the holding capacity significantly. Without a second precontraction in the test position, participants were only able to reach ~53% of the AF_max_ under isometric conditions. Hence, AF_max_ was then reached during muscle lengthening. Only one participant was able to reach ~100% of AF_max_; thus, he did not yield during the MMT after procedure CL. In contrast, for MMTs after the short and low-intensity second contraction in the test position (CL-CT), as well as for regular MMT, the participants were able to reach more than 98% of the AF_max_. Overall, 16 of 19 limbs were able to generate 100% of the AF_max_. Hence, they reached their maximal strength under isometric conditions. The remaining three limbs still showed considerably high ratios (95.67–99.97%), except for one with a ratio of only 77.46%. The latter clearly lengthened the muscle during the MMT after procedure CL-CT. Overall, the second precontraction in test position led to an instantaneous regaining of regular holding capacity; thus, the musculoskeletal instability could be resolved. The objective data support the subjective ratings of the tester in all cases. All regular MMTs were rated as “stable” by the tester. For procedure CL, the tester assessed two of 38 MMTs as “stable” (belonging to the same participant) and 36 as “unstable”, whereas, with procedure CL-CT, 33 of 37 MMTs were rated as “stable”, and four were rated as “unstable” (belonging to the same participant).

### 3.3. Onset of Oscillations in the Course of Adaptive Force Comparing the Different Procedures

RM ANOVA showed a significant main effect (Table 2), whereby pairwise comparisons revealed a significantly higher AFosc for procedure CL compared to regular MMT (*p*_adj_ = 0.001, *d_z_* = 0.998) and to procedure CL-CT (*p*_adj_ < 0.001, *d_z_* = 1.434), respectively. Regular MMT and procedure CL-CT did not differ significantly (*p*_adj_ = 1.000) (Figure 3c).

The ratios regarding the onset of oscillations made the results even clearer; the upswing of oscillations started on a −20.66% and −14.66% lower level in relation to AF_max_ for regular MMT and procedure CL-CT, respectively (Table 2, Figure 3e). Furthermore, the oscillatory upswing occurred under isometric conditions (ratio AFosc/AFiso_max_) for regular MMT and procedure CL-CT (stable MMTs), whereas, for procedure CL (unstable MMTs), oscillations did not occur or arose during the lengthening phase (Figure 3f). Only in one case the oscillations did arise at 95% of AFiso_max_, thus, still under isometric conditions. These findings indicate that the occurrence of oscillations might be a prerequisite for adequate adaptation and, hence, for musculoskeletal stability.

### 3.4. Maximal Voluntary Isometric Contraction

The MVIC amounted initially to 275.02 ± 46.66 N (range: 204.87–356.13 N) and 265.78 ± 47.55 N (range: 202.67–348.74 N) on average after the six MMT trials. Thus, the MVIC declined by ~3.36% which was marginally nonsignificant (t (18) = 2.103, *p* = 0.050). Hence, a significant fatiguing effect was not present. AFiso_max_ was statistically similar to MVIC for stable MMTs (regular MMT: −0.51% ± 7.69%; CL-CT: −4.21% ± 14.75%); after procedure CL, it was clearly lower (−45.05% ± 25.51%). These findings indicate that, under stable conditions, the maximal isometric forces were similar for holding and pushing actions in the present setting. This indicates that the tester applied adequate forces during the MMTs.

## 4. Discussion

The study investigated the effects of two different preceding procedures—assumed to manipulate muscle spindles—on AF parameters. There are two central results of the study. Firstly, after the CL procedure, the maximal holding capacity (AFiso_max_) was almost halved compared to regular MMT and MVIC. Secondly, this reduction did not appear when a second short contraction with ~25% of MVIC was added in test position immediately prior to the test. Consequently, during the CL-CT procedure, the muscle behaved in the same way as for regular MMT. Moreover, the maximal AF was not affected by the preceding procedures. Hence, all hypotheses were verified positively.

### 4.1. Methodological Considerations Regarding the Comparison of the Procedures

From the methodological point of view, it was important to assess the force level at which the MMTs were executed. The reached maximum depends on the interaction of both partners. In case the force produced by the tester is too low, an MMT falsely assessed as stable could be the result even though the tested muscle would be actually unstable. Therefore, the reached maximal forces must be taken into consideration. Compared to regular MMTs, AF_max_ was 2.5% higher after CL and −2.7% lower after CL-CT. Accordingly, the developed forces were really similar. Therefore, an inappropriate maximal force produced by the tester can be ruled out for the found differences. Moreover, the forces during all MMTs reached the level of the initial MVIC. The AF_max_ during regular, CL, and CL-CT procedures amounted to 99.77%, 102.36%, and 97.07% of the MVIC, respectively. Therefore, all MMTs ran with at least approximately maximum intensities. In addition, the slope of force development should be taken into account. A fast increase would make it more difficult for the tested muscle to lock into a stable resistance (for neurophysiological considerations, see [35]). The slope of regular MMT was significantly lower compared to the MMTs after both procedures, CL and CL-CT. However, it is assumed that, in this study, this had no relevant influence on the further outcomes because the regular MMT and CL-CT showed similar AF parameters despite differing slopes. Furthermore, the slopes matched between CL and CL-CT, but AFiso_max_ and AFosc differed significantly.

Another prerequisite to compare both procedures was similar precontractions with regard to temporal sequence and intensity. No technical feedback was applied to avoid distractions so that the participant could fully concentrate on the sensations of the arm during the tests. Although the timing of both procedures (CL and CL-CT) was based solely on the temporal feeling of the tester, the duration of CL precontractions was very close to the instructed 1 s (±0.1 s). It differed only by 3/100 s on average between both procedures. In addition, the time period between CL precontraction and the start of MMT was also controlled by the commands of the tester. He had to temporize and bridge the duration of the additional second precontraction of CL-CT during the CL procedure. Because this time period also showed a small (0.18 s) and nonsignificant difference, the temporal sequences could be rated as similar. In contrast, the duration of the second contraction during CL-CT was substantially shorter than instructed (0.72 s instead of 1 s). Furthermore, the intensities of the precontractions were mostly underestimated by the participants. Instead of the instructed 20% of the MVIC, the participants applied a higher intensity by at least 10% in 65% of all precontractions; in 18% of cases, the intensity was lower than predefined. In the remaining 18%, the self-estimated intensity of precontraction was within 18–22% of the MVIC. Overall, the second precontraction during CL-CT was performed more quickly but more intensely than given. The mentioned differences do not indicate an essential interference of the planned design but should be taken into account in the later discussion.

### 4.2. Neurophysiological Considerations

The two main results mentioned above are in line with the findings of Gregory et al. that, after a procedure similar to CL the stretch reflex was reduced and could be resolved by a second precontraction in the test position [18]. They explained the inhibitory effect found after a precontraction in lengthened position followed by a passive muscle shortening (CL procedure refers to “hold-long” in Gregory et al.) by a generated slack of the intrafusal fibers [18]. It has been shown early that muscle spindles can fall slack or taut depending on the preceding history of muscle length and contraction [16]. A slack appears with CL conditioning during the subsequent passive shortening. There seems to be specific kinds of muscle spindles which fall silent under this condition, as shown in cats [21,43]. It was suggested that muscle spindles are unable to maintain their resting discharge under slacked conditions [43]. It is assumed here that—in this case—the spindle afferents were not adjusted to the shortened overall muscle length at the test position. The slacked fibers were not able to provide appropriate information on length and lengthening of the muscle being tested. According to suggestions by several authors [19,21,44,45,46,47,48], the short additional precontraction in the test position fixed this misalignment during CL-CT, which then allowed an appropriate stabilization during the following MMT. Obviously, the second precontraction here was sufficient to resolve slack although it was significantly shorter than the first precontractions. This was indicated by the regained level of AFiso_max_ approximately to the amount of AF_max_. Presumably, not the duration but the intensity of the precontraction is the main factor to remove the slack in muscle spindles; as mentioned in Section 1, at least 5% of the MVIC, partly up to 10%, is assumed to be sufficient [18,20]. Basically, the full range of mechanoreceptors of muscles, tendons, capsules, connective tissue, and skin is involved in each motion and contraction. However, the CL procedure is likely to influence mostly muscle spindles. This suggests that the observed phenomena may mainly be due to those receptors.

It seems comprehensible that a slacked muscle spindle can affect a quick stretch reflex. In contrast, the present study showed a reduced holding function during the whole period of an MMT, which lasted around 3 s. In CL-CT, a brief contraction immediately before the MMT revoked that destabilizing effect. It has to be questioned why the muscle contraction during the MMT after the CL procedure did not have the same effect. It seems that the condition at the start of the MMT influences the whole process in the course of external force increase. The role of the participant can be understood as a slave system which has to execute a follow-up control in response to a varying reference input, as also applied in technical motion control systems. It is hypothesized that the initial state of proprioceptive inputs influences the iterative processes of the adaptive control loop during the whole process of a running test. (It is likely that a subsequent second MMT would be stable. However, this was not considered in the present study).

According to this hypothesis, a complete holding muscle action could be understood as a prolonged reflex sequence. A sufficient muscular stabilization would be based, inter alia, on the appropriate initial adjustment of the involved muscle spindles. Preceding actions in everyday motions or sports which accidentally led to a slack could impair stability during a single motion. Such a process could have an effect on motor actions which acquire stability, e.g., stopping a motion, changing direction, or compensating for external impact in sports. Muscle strain injuries are especially related to decelerating actions [49,50]. Sports injuries appear very often during movements which are executed 1000 times without any problem. Hence, there must be a specific condition during the one motion in which the injury occurs. Preceding actions which impair muscle spindles could be one trigger.

The results showed a high interindividual variability regarding the reduction in AFiso_max_—interpreted as reduced stability—for the CL procedure (CV = 0.517). As Figure 4 shows, the residual holding force ranged from 10% to 100% with regard to AF_max_. The individual behaviors varied largely. In four cases, the decrease went below 40% of AF_max_ for both trials after the CL procedure (Figure 4: limbs no. 3, 4, 6, and 15). One muscle (limb no. 19) showed no reaction to the CL procedure and stayed completely stable irrespective of the conditioning. Another participant (limbs no. 8 and 9) developed a reduction in AFiso_max_ during the CL procedure (~54.5% of AF_max_ for first and ~52.8% for second measured side averaged over both MMTs) but did not return to full stability during CL-CT (~90.8% of AF_max_ for first and ~77.5% for second measured side). In this case, the revoking effect of the second precontraction obviously was not as high as for the other participants. AFiso_max_ after CL still amounted to only ~64% of AFiso_max_ after CL-CT. Additional trials (not included in this study) showed that, in this particular participant, a second contraction in the test position with predefined self-estimated 50% of the MVIC was necessary to erase the inhibitory effect of a preceding CL procedure. The broad variety of reactions shows once more the individuality of the complex motor control. The afferent input of the spindles of a single muscle is only one voice in a true choir of proprioception—although likely the most important one. In contrast to monosynaptic reflexes, the voluntary action during an MMT runs not only on a spinal level but also on a supraspinal one, where numerous additional influences can modulate the motor action. The mathematical model of Blum et al. [14] showed that muscle spindles must be understood as part of a complex multiscale biophysical framework including tendon organs.

A further aspect regarding the complex neuromuscular control circuitries is the oscillations appearing during interpersonal muscle actions [39,51,52,53]. This observed oscillatory synchronization presumably requires undisturbed functionality of the involved neuromuscular systems including proprioception. The MMT is a special kind of muscular interaction between two persons. It was found here that, when muscles showed high stability (regular MMT and CL-CT), the muscular interaction was accompanied by an early onset of oscillations at ~80% of AF_max_. This characteristic mostly disappeared when muscles became unstable after the CL procedure. The combination of a high AFiso_max_ with an early upswing of mutual oscillations of both involved partners was already found in previous studies as a sign of muscular stability [30,31,32,34]. When muscles lose this property, the partners seem to be unable to find a common muscular rhythm. The results of the present study suggest that appropriate adjusted muscle spindles could be one necessary condition linking the two interacting neuromuscular systems to generate mutual oscillation. With the CL procedure, the presumed impairment of spindles occurred on the side of the participant, whereas the tester was not influenced. It remains unclear whether a manipulation on the tester’s side would have also led to missing or late oscillations. It is assumed that the side of the participants could be the decisive one because they perform the HIMA, which includes the reaction to the external force given by the pushing tester. The frequency of such interpersonal oscillations is ~10 Hz [39,51,52,53]. The 10 Hz rhythm in muscles is interpreted as physiological tremor, which presumably originates from multiple factors including reflex loop resonances [54]. The long latency reflex lasts ~100 ms reflecting the latency of proprioceptive signals [55,56]. Such low-frequency oscillations are also found in different brain areas which participate in motor control, such as the inferior olivary nucleus, the cerebellum, and the thalamus [57,58,59]. Impairing muscle spindles by generating a slack seems to impede the ability to create a common interpersonal rhythm and to reduce the ability to adapt to an external force in a holding manner. This highlights the importance of undisturbed muscle spindle afferents for musculoskeletal stability and neuromuscular functioning.

### 4.3. Limitations

The performed measurements were based on a manual muscle test, which is subjective by nature. Using a handheld device, the MMTs could be objectified. With respect to scientific quality criteria, it is necessary to point out that the tester’s ability to generate reproducible force profiles was proven in a previous study [35]. All methodological aspects of the measured biomechanical parameters in the comparison of different procedures were given and discussed above, and they can be rated as suitable. Despite the low number of included participants, the clear results gave an unambiguous picture. Nevertheless, the findings should be verified in future studies with a greater number of participants.

## 5. Conclusions

The adaptive holding muscle function reacted to a generated slack in muscle spindles with a significant reduction. This suggests that holding against (varying) external forces requires appropriate proprioception—especially regarding length and tension control—and likely contains reflex components. Since the maximal AF was not affected by the different procedures, the holding isometric Adaptive Force (AFiso_max_) should be considered as an independent type of muscle action mirroring interference in the complex control circuitries more sensitive than the usual measurements of the MVIC or other pushing forces.

Since a reduced holding function is interpreted as musculoskeletal instability, this special muscle function should be regarded with respect to risk of injury and to the development of musculoskeletal complaints in sports or everyday activities where adaptation of force to the external circumstances is almost always necessary.

It is concluded that assessing the holding capacity could be a useful approach to support diagnostics to investigate the functional state of the neuromuscular system, which offers advantages over conventional measurements of maximal forces.

Eventually, of course, further investigations are needed to clarify whether there are specific motions in everyday activities or sports with a certain risk to impair muscle spindles and, therefore, the quality of motion. The considered procedures including AF measurements could also be an approach to assess the processing of muscle spindle afferents in motor control in subjects with specific conditions such as aging and pathologies or during convalescence after muscle injuries.

## Figures and Tables

**Figure 1 life-13-00911-f001:**
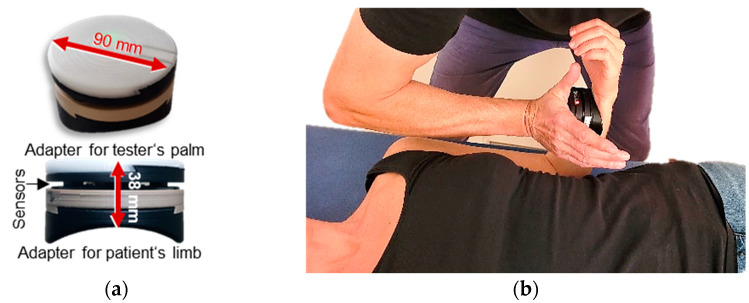
Equipment and setting: (**a**) handheld device; (**b**) starting (test) position of the manual muscle test and MVIC test of elbow flexors (modified according to [30,32]).

**Figure 2 life-13-00911-f002:**
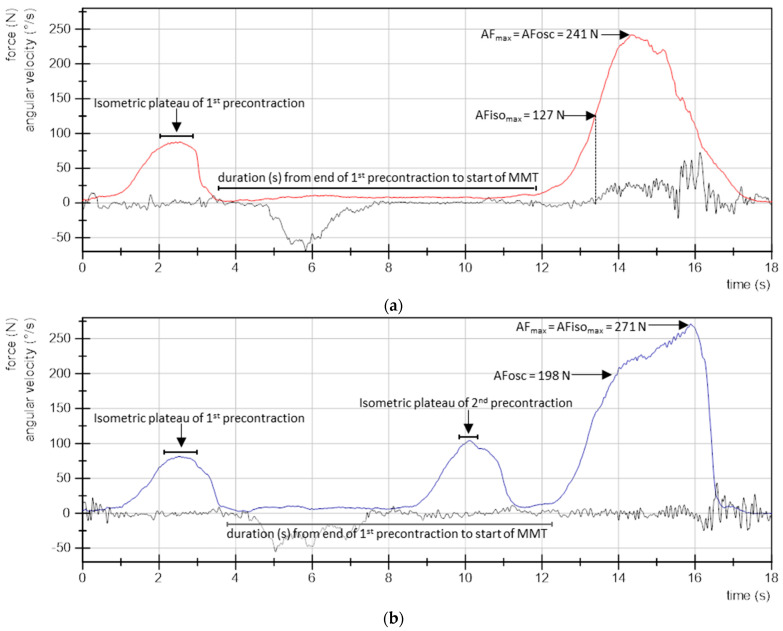
Curve characteristics. Exemplary curves of force (N, red and blue curves) and angular velocity (°/s, black curves) of (**a**) one MMT after procedure CL (precontraction in lengthened position with passive return), and (**b**) one MMT after procedure CL-CT (CL with a second precontraction in test position). The evaluated parameters are indicated: maximal Adaptive Force (AF_max_); maximal isometric AF (AFiso_max_); AF at onset of oscillations (AFosc); isometric plateaus of precontractions for calculation of the averaged force and duration, as well as the duration from end of first precontraction to start of MMT.

**Figure 3 life-13-00911-f003:**
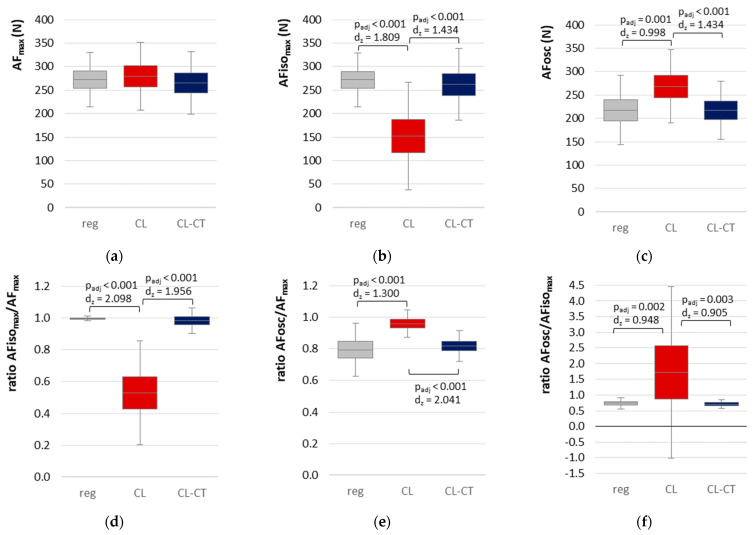
The 95% confidence intervals including arithmetic mean and standard deviation (error bars) comparing regular MMT (reg, gray), MMT after procedure CL (red), and MMT after procedure CL-CT (blue) for the different parameters are displayed: (**a**) maximal Adaptive Force (AF_max_), (**b**) maximal isometric AF (AFiso_max_), (**c**) AF at onset of oscillations (AFosc), (**d**) ratio AFiso_max_ to AF_max_, (**e**) ratio AFosc to AF_max_, and (**f**) ratio AFosc to AFiso_max_. Adjusted *p*-values of pairwise comparisons (Bonferroni corrected) and effect sizes Cohen’s *d_z_* are given in case of significance.

**Figure 4 life-13-00911-f004:**
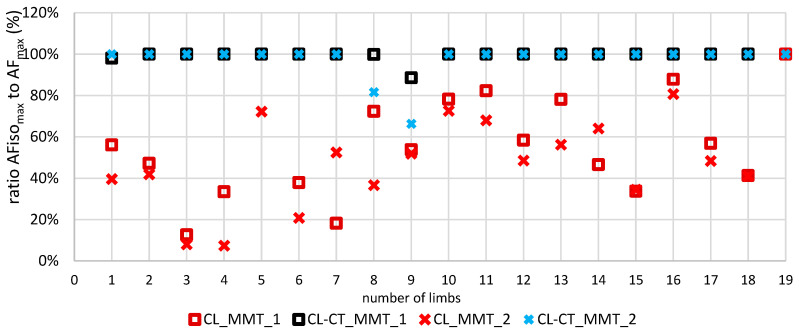
Single values of the ratio AFiso_max_ to AF_max_ (%) for the first (MMT_1, square) and second (MMT_2, cross) trials of procedure CL (red) and procedure CL-CT (blue) for each of the 19 tested limbs (elbow flexors).

**Table 1 life-13-00911-t001:** Precontractions. Arithmetic means ± standard deviations of the proportion of force of pre-contraction related to the individual’s MVIC (%), the duration of precontraction phases (s), and the duration from the first precontraction to the start of MMT (s) for procedures CL and CL-CT.

Parameters	Procedure CL	Procedure CL-CT
Force of first precontraction/MVIC (%)	25.04 ± 9.44	26.44 ± 9.11
Force of second precontraction/MVIC (%)	-	28.47 ± 8.81
Duration of first precontraction (s)	1.09 ± 0.33	1.12 ± 0.33
Duration of second precontraction (s)	-	0.72 ± 0.12
Duration of first precontraction to MMT (s)	7.51 s ± 0.96	7.33 ± 1.76

MMT = manual muscle test; CL = precontraction in lengthened position with passive return; CL-CT = CL with second subsequent precontraction in test position.

**Table 2 life-13-00911-t002:** Arithmetic means (M) and standard deviations (SD) of the maximal Adaptive Force (AF_max_), the maximal isometric AF (AFiso_max_), and the AF at onset of oscillations (AFosc) (all in N), as well as the respective ratios and slope [lg(N/s)] for the regular MMT, MMT after procedure CL, and MMT after procedure CL-CT. Statistical values (F value, degrees of freedom (df), significance *p*, and effect size η^2^) of RM ANOVA are given. Superscript values indicate significant pairwise comparisons.

Parameter	Procedure	M	SD	F	df	*p*	η^2^
AF_max_	Regular	272.574	38.846	3.193	2, 36	0.053	-
	CL	279.512	49.427				
	CL-CT	265.259	46.158				
AFiso_max_	Regular	271.774	39.367	44.946 *	1.24, 22.36	<0.001 ^1^	0.714
	CL	152.146	78.607				
	CL-CT	262.033	52.597				
AFosc	Regular	217.580	51.058	18.992	2, 36	<0.001 ^1^	0.512
	CL	268.585	53.937				
	CL-CT	217.255	43.088				
Ratio AFiso_max_/AF_max_	Regular	0.997	0.100	75.660	1.09, 19.64	<0.001 ^1^	0.808
	CL	0.530	0.225				
	CL-CT	0.983	0.055				
Ratio AFosc/AF_max_	Regular	0.795	0.115	26.644 *	1.47, 26.54	<0.001 ^1^	0.597
	CL	0.959	0.059				
	CL-CT	0.818	0.066				
Ratio AFosc/AFiso_max_	Regular	0.797	0.119	16.264 *	1.01, 18.12	0.001 ^1^	0.475
	CL	2.566	1.885				
	CL-CT	0.836	0.089				
Slope	Regular	2.017	0.140	19.686	2, 34	<0.001 ^2^	0.537
	CL	2.210	0.201				
	CL-CT	2.137	0.121				

* Greenhouse–Geisser correction. ^1^ Significant pairwise comparisons for CL vs. CL-CT and CL vs. regular MMT. ^2^ Significant pairwise comparison for regular MMT vs. CL and MMT vs. CL-CT.

## Data Availability

The data presented in this study are available in the article.

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
