# Peer review of "How to Confuse Motor Control: Passive Muscle Shortening after Contraction in Lengthened Position Reduces the Muscular Holding Stability in the Sense of Adaptive Force"

_life, 2023, doi:10.3390/life13040911_

Round 1
Reviewer 1 Report
In this study, authors investigated the influence of consecutive static force exertion at different elbow flexion. The study is interesting and well written. The methods are clearly stated, even if not fully replicable because of the presence of a human experienced tester. However, this limitation is properly stated in a paragraph in the conclusions.
Following are some minor comments that I think would improve your paper.
Line 162: From the results, it seems that what you identified with ‘19 muscles’ are actually the number of the tested limbs (in some participants a single limb was tested, in other participants both were tested). I think this could be confounding because multiple muscles act as elbow flexors, so it may looks that you collected the activity form 19 muscles. So I suggest to use another expression to define the ‘number of limbs’ you tested (e.g. number of ‘testing limb’, or number of ‘tests’).
Line 180: Usually isometric tasks imply that the posture is completely fixed, i.e. because the end-effector is rigidly connected to the setup. In the proposed protocol, as the posture is kept constant not because of a rigid connection, I would call this a ‘static’ rather than ‘isometric’ task.
Line 194: Did you compared the stability assessment of the tester with the data of the angular velocity acquired with the device to provide an objective measurement whether the trial was ‘stable’ or not?
Author Response
Dear Reviewer,
thank you for your positive reply, interest in our research and helpful comments.
We addressed all of them – see point-by-point-response below.
Thank you for your effort!
Sincerely,
the authors
Point-by-point-response:
Line 162: From the results, it seems that what you identified with ‘19 muscles’ are actually the number of the tested limbs (in some participants a single limb was tested, in other participants both were tested). I think this could be confounding because multiple muscles act as elbow flexors, so it may looks that you collected the activity form 19 muscles. So I suggest to use another expression to define the ‘number of limbs’ you tested (e.g. number of ‘testing limb’, or number of ‘tests’).
- Thank you for that hint. We agree that this might be confusing. We changed it to “limbs” (in Line 162 and in Lines 365-389 inkl. Figure 4) and in discussion lines 513 ff)
Line 180: Usually isometric tasks imply that the posture is completely fixed, i.e. because the end-effector is rigidly connected to the setup. In the proposed protocol, as the posture is kept constant not because of a rigid connection, I would call this a ‘static’ rather than ‘isometric’ task.
- This is an interesting point. And again, you are correct. The position was static, the muscle action still was isometric – since muscle length stayed similar. We changed it accordingly – if we spoke of position, we used the term “static”. In case it refers to the muscle action, we left “isometric” since the muscle actions (AF and holding) was termed that way…we hope this is in line with your suggestion.
Line 194: Did you compared the stability assessment of the tester with the data of the angular velocity acquired with the device to provide an objective measurement whether the trial was ‘stable’ or not?
- Yes we did. All MVIC test were ‘stable’ according to angular velocity. We added this in the method section (Lines 253-254): “According to the gyrometer signals all MVIC tests were conducted under static conditions.“
Thank you for that hint!
Reviewer 2 Report
I congratulate the authors on a very successful study. The entire manuscript is carefully processed.
Introduction – is processed in a satisfactory manner, the authors logically classify the topics that they introduce into the problem.
Material and methods – all methodological procedures are well developed and this passage cannot be faulted. The only problem is the small number of probands, but the authors are aware of this fact and state it within the limits.
Results – this passage is also processed logically, with great precision.
Discussion – this passage is quite extensive, it is systematically structured, and its content is a fully satisfactory solution to the entire thesis.
Conclusion - I have only one comment about this passage, and that is that it is too extensive. It would be appropriate to shorten this subchapter and describe the conclusions more concisely. The authors argue unnecessarily, in some places this passage comes close to discussion.
Minor notes: - Table 1 – there should not be a parenthesis after the abbreviation CL - It would be convenient if the entire table were expenses on one side. - It would be appropriate to justify in the text why such a low number of probands was used.
The results of the conducted study indicate and can be said to confirm the considered assumption that the sensitivity of the muscle spindle plays an important role for neuromuscular functioning and musculoskeletal stability. This insight is very important for practice, and despite the fact that it is generally considered, this study is one of the few that attempts to prove this fact. I definitely recommend the manuscript for publication.
Author Response
Dear Reviewer,
Thank you for your very positive reply and interest in our research. We really appreciate the openness for such new approaches.
Thank you for your hints in Table 1 (we deleted the parenthesis) and extended Table 2 on one page (this will be re-formatted anyway before publication).
The sample size was justified in section “participants”:
“G*power (version 3.1.9.7, Düsseldorf, Germany) was used for a priori sample size estimation. For the main comparison regarding the different procedures (regular MMT vs. CL vs. CL-CT procedures) RM ANOVA within factors was chosen (α = 0.05, 1–β = 0.8, correlation: 0.5, non-sphericity correction: 1, number of groups: 1, number of measurements: 3). To determine a substantial effect size of f = 0.4 a minimum of n = 12 participants was revealed. According to previous studies investigating AF parameters by MMTs [30–32], the effect sizes were very large so that the here estimated effect size of 0.4 can be considered conservative.”
Additionally, we shortened the conclusion (from 356 to 236 words) – we are aware that we tend to write extensive conclusions. In case you prefer an even shorter version, we would reduce it further.
We hope that we could address your concerns by the revisions made.
Thank you for your effort!
Sincerely,
The authors